# HPV and Cervical Cancer: A Review of Epidemiology and Screening Uptake in the UK

**DOI:** 10.3390/pathogens12020298

**Published:** 2023-02-11

**Authors:** Sunyoung Choi, Ayden Ismail, George Pappas-Gogos, Stergios Boussios

**Affiliations:** 1GKT School of Medicine, King’s College London, London SE1 9RT, UK; 2Department of General Surgery, University Hospital of Alexandroupolis, School of Medicine, Democritus University of Thrace, 6th Kilometer, 68100 Alexandroupolis, Greece; 3Department of Medical Oncology, Medway NHS Foundation Trust, Windmill Road, Kent, Gillingham ME7 5NY, UK; 4Faculty of Life Sciences & Medicine, School of Cancer & Pharmaceutical Sciences, King’s College London, London SE1 9RT, UK; 5Kent Medway Medical School, University of Kent, Kent, Canterbury CT2 7LX, UK; 6AELIA Organization, 9th Km Thessaloniki—Thermi, 57001 Thessaloniki, Greece

**Keywords:** HPV, cervical cancer, epidemiology, risk factors, cervical cancer screening, HPV vaccination

## Abstract

Cervical cancer is the fourth most common malignancy in females worldwide, and a leading cause of death in the United Kingdom (UK). The human papillomavirus (HPV) is the strongest risk factor for developing cervical intraepithelial neoplasia and cancer. Across the UK, the national HPV immunisation programme, introduced in 2008, has been successful in protecting against HPV-related infections. Furthermore, the National Health Service (NHS) implemented the cytology-based cervical cancer screening service to all females aged 25 to 64, which has observed a decline in cervical cancer incidence. In the UK, there has been an overall decline in age-appropriate coverage since April 2010. In 2019, the COVID-19 pandemic disrupted NHS cancer screening and immunisation programmes, leading to a 6.8% decreased uptake of cervical cancer screening from the previous year. Engagement with screening has also been associated with social deprivation. In England, incidence rates of cervical cancer were reported to be 65% higher in the most deprived areas compared to the least, with lifestyle factors such as cigarette consumption contributing to 21% of cervical cancer cases. In this article, we provide an update on the epidemiology of cervical cancer, and HPV pathogenesis and transmission, along with the current prevention programmes within the NHS.

## 1. Introduction

Cervical cancer is a highly prevalent disease amongst females, associated with significant morbidity and mortality worldwide [1]. It is the fourth most common malignancy to affect women globally and responsible for approximately 850 deaths per annum in the United Kingdom (UK) [2,3]. Cancer Research UK reported 3200 new cervical cancer cases in the UK annually between 2016 and 2018, with peak incidence rates among females aged 30 to 34 [3]. There are several associated risk factors, with the human papillomavirus (HPV) infection being a major causative factor, contributing to approximately 99.7% of all cases [4]. Other factors have also been reported to increase cervical cancer risk, including smoking, immunosuppression, poor sexual health, and screening non-attendance [5]. Regardless of the advances in the primary and secondary prevention of cervical cancer, a significant portion of patients present with or develop metastatic disease, mainly within the first 2 years of completing primary treatment. Surgical resection and/or radiotherapy may potentially be curative for selected cases with locally recurrent or limited metastatic disease [6]. However, for the vast majority of patients, palliative chemotherapy represents the only treatment. Yet, there is a significant unmet clinical need for effective second-line therapeutic strategies [7]. Technologies of proteomics, such as mass spectrometry and protein array analysis, have advanced the dissection of the underlying molecular signaling events in gynaecological oncology [8]. Moreover, proteomics analysis of cervical cancer can uncover new therapeutic targets, which can reduce the emergence of drug resistance and improve the prognosis [9]. This review aims to address the epidemiology of cervical cancer, HPV pathogenesis and transmission, and current programmes within the National Health Service (NHS) for prevention and early identification of this disease. Detailed analysis into screening uptake, risk factors, and future developments are crucial in reducing the burden of cervical cancer in the UK.

## 2. Epidemiology of Cervical Cancer

Cervical cancer is a devastating malignancy of the cervix, with squamous cell carcinomas reported to be more prevalent than adenocarcinomas [10,11]. The year 2020 recorded an estimated 604,000 new cases of cervical cancer and 324,000 deaths globally, with almost 90% of these cases occurring in low- and middle-income countries [2,12]. The NHS first aimed to tackle the burden of this disease by introducing a national cervical screening programme in 1988, which has since seen a significant reduction in over a third of cases in England [13]. Cervical cancer screening is available from the age of 25, as the disease is rare among younger individuals [14]. A large multi-centre study in the UK reported that screening individuals between the ages of 20 to 24 years old had little or no effect on the rates of cervical cancer [15]. Interestingly, 9% of new diagnoses are in those aged 75 years and older, but this demographic is not currently screened as part of the programme [14]. The screening programme invites all people with a cervix between the ages of 25 and 64, to undergo screening for cervical cancer every 3 or 5 years, depending on age [16]. Since then, the incidence rates have remained stable over the last decade [14]. Moreover, incidence rates of cervical cancer in England have been compared with countries such as Portugal, exploring differences between organised and opportunistic screening methods on cervical cancer incidence and mortality [17]. Despite both England and Portugal observing a decline in mortality, Portugal adopts an opportunistic screening programme that has reported higher rates of cervical cancer cases compared to organised screening methods in England [17].

In the UK, it is important to note that cases have been reported to be 65% higher in areas of deprivation [14]. There are several risk factors associated with this malignancy, HPV infection identifying as the strongest causative factor [18]. HPV types, particularly HPV 16 and 18, are found to be highly prevalent in the UK general population [19]. The HPV is a sexually transmitted infection that increases the risk of an individual to develop cervical cancer [20]. Multiple sexual partners, early sexual intercourse, and smoking are established behavioural risk factors for developing cervical cancer, which may be more commonly seen in areas of higher social deprivation [21]. Other than sexual transmission, other modes of disease conveyance include vertical transmission from mother to infant and through fomites, such as clothing, although the role of fomites in the transmission of infection is not fully understood [22]. In May 2018, the World Health Organisation (WHO) proposed the ‘Cervical Cancer Elimination Initiative’, which aims to eradicate cervical cancer globally through more rigorous vaccination and screening methods [23]. As HPV-associated malignancy may be preventable, understanding the pathogenesis of the HPV infection in developing invasive cancers will form the basis of preventing and treating most cases, thus, significantly improving the outcome of these patients.

## 3. Introduction of HPV

### 3.1. HPV Subtypes

HPVs belong to the Papillomaviridae family, which are small, non-enveloped, double-stranded DNA viruses, and the most common viral culprit of reproductive tract infections [12,24,25]. The genome of HPV is circular and estimated to be eight kilobase pairs in size, with most encoding eight proteins that can be divided into either ‘early’ or ‘late’ regions [25]. The ‘early’ region contains six proteins (E1, E2, E4, E5, E6, and E7), which are responsible for HPV genome replication and immune modulation [25,26]. The ‘late’ region consists of L1 and L2 capsid proteins, which have been reported to play a role in the transmission of the virus [24]. Not all HPV types have oncogenic potential, with more than 200 HPV subtypes identifiable as either low or high-risk based on their degree of oncogenicity [25,26,27]. Low-risk HPVs, such as HPV 6, 11, 42, 43, and 44, are classified as such due to commonly only causing benign epithelial lesions, such as verrucae, warts, and papillomas [25,27]. On the other hand, high-risk HPVs (hrHPV), including HPV 16, 18, 31, 33, 34, 35, 39, 45, 51, 52, 56, 58, 59, 66, 68, and 70, are strongly linked to the development of cervical, anal, penile, vulval, vaginal, and oropharyngeal cancers [25,27]. Approximately 40 HPVs are specifically related to the anogenital region, with HPV 16 and 18 considered the most potent hrHPVs, responsible for almost half of high-grade cervical pre-malignancies [12]. Specifically in the UK, a 2014 study reported that hrHPV types 16 and 18 were highly prevalent (83%) in women under the age of 30, whilst other hrHPV types (31, 33, 45, 52, and 58) were detected in 16.1% of hrHPV-positive cervical cancers regardless of age [19].

### 3.2. HPV Transmission

HPV is primarily spread through direct skin-to-skin or skin-to-mucosa contact [28]. Sexual activity (including vaginal, anal, or oral intercourse) with an individual who has active HPV infection, is the most common mode of transmission [29]. Additionally, there is non-sexual horizontal transmission of HPV through skin, mouth, or fomite contact, which are less common [28,30]. Regarding cervical cancer, some risk factors are associated with its sexually transmitted nature, such as sexual intercourse from an early age and having multiple sexual partners [31,32]. HPV infection is highly prevalent, with one study estimating that women in the United States (US), who have similar interventions to the UK, carry a lifetime risk of infection between 53.6% and 95%, thus, making it a priority to reduce the burden of this disease [33].

## 4. Pathophysiology of HPV-Related Malignancy

Ninety percent of HPV-induced cell changes in the cervix regress spontaneously and HPV is cleared from the body, without any oncogenic consequences [27]. However, when a hrHPV infection evades the immune system, which is particularly common among immunocompromised patients, dyskaryosis and cervical intraepithelial neoplasia (CIN) can occur [34]. CIN is a pre-malignant abnormal collection of cells that, if untreated, may progress to become invasive carcinoma [35]. The pathophysiology is not fully understood, but it is believed that micro-abrasions in the epithelial surface are the most probable route for HPVs to infect the epithelial basal layer [24,36]. After the interaction of the HPV viral DNA with the host cell, the virus is then internalised and transported into the nucleus for further replication [37,38]. E6 and E7 are proteins that possess strong oncogenic potential by interacting with tumour suppressor genes (TSGs), the tumour protein p53 (TP53), and retinoblastoma proteins (pRb), respectively [39,40]. E6 targets TP53 for degradation via the ubiquitin pathway, thus, preventing apoptosis [39,40]. E7 targets the retinoblastoma family members RB1, RBL1, and RBL2, which, consequently, drives the oncogenic process [39,41]. HPV rely on the replication system of the human cell, as they do not have their own. Accordingly, the intrinsically differentiated dormant cells must be induced to replicate again, which the viruses achieve with E6 and E7 activity. Additionally, E5 is considered a weak co-factor that can accelerate oncogene expression, although any significant effects on E6 and E7 are yet to be proven [42]. Therefore, downregulation of these TSGs causes genomic instability, thus, driving the malignant transformation of an HPV-infected cell into an invasive cancer cell.

Basal epithelial cells have intimately related replication cycles to HPV, whereby early gene expression results in substantial amplification of viral DNA [43]. In the epithelial midzone and superficial zones, further viral replication of the genome and expression of L1, L2, and E4 genes take place [24,39]. L1 and L2 enclose the viral genomes to form progeny virions in the nucleus, which then go on to form new infections [39,44]. Due to the non-enveloped nature of the virus, the capsid has a non-lipid membrane structure, in which L1 plays a vital role in its formation [45,46]. Therefore, targeting L1 with self-assembled virus-like particles (VLPs) has become an integral part of developing successful HPV vaccines [45,46].

## 5. Other Risk Factors

### 5.1. Smoking

Whilst the relationship between HPV infection and cervical cancer is well-established, it is important to acknowledge other possible risk factors. Both squamous cell and adenocarcinoma of the cervix have similar risk factors, with cigarette smoking being a particularly strong risk factor for squamous cell carcinomas [47]. The large European Prospective Investigation into Cancer and Nutrition (EPIC) study recruited over 521,000 participants across 10 European countries, which revealed a significant two-fold increase in CIN and invasive cervical cancer risk with greater pack-years of smoking, whilst time since quitting reduced risk by two-fold [48]. The association between smoking and cervical cancer risk has been debated, but it is thought to have a multi-faceted role in cervical carcinogenesis [49]. Smoking has been thought to either increase the risk of HPV infection or decrease the clearance, leading to an increased risk of cervical malignancy [50,51]. One study discovered that benzo[a]pyrene, a carcinogenic component of cigarettes, enhanced HPV persistence, thus, increasing HPV 16 and 18 titres [52]. Other studies have supported the hypothesis that smoking promoted carcinogenesis by suppressing cell-mediated immunity against HPV infection [53,54]. Therefore, the pathophysiology of cigarette smoking in cervical carcinogenesis seems to be a multifactorial mechanism that requires further research to understand the complexity of this phenomenon.

### 5.2. Oral Contraceptive Pill

The oral contraceptive pill (OCP) is the most common form of contraceptive alongside barrier protection in the UK, due to its high efficacy [55]. Whilst the National Institute for Health and Care Excellence (NICE) states that OCP use increases cervical cancer risk, the association is not well-established and has been the subject of debate among clinicians [56,57]. Several systematic reviews and meta-analyses concluded that there was no evidence to suggest that OCP use was associated with increased risk, whilst others concluded that hormonal contraceptives have the potential to affect the HPV-dependent pathway of carcinogenesis due to the cervix being an oestrogen-sensitive organ [58,59,60,61,62]. A recent systematic review and meta-analysis found that OCP use is significantly associated with cervical cancer, particularly adenocarcinomas [57]. Perhaps individuals taking the OCP may be less likely to use concomitant barrier protection, thus, becoming more susceptible to acquiring the HPV infection. Several longitudinal studies found that barrier methods, such as condom use, showed a statistically significant protective effect against the HPV infection and CIN development [63,64,65]. The International Agency for Research on Cancer (IARC) announced that the OCP was a risk factor for developing cervical cancer, stating the increased risk is due to long-term use of more than 5 years [66]. More research into the possible carcinogenic properties of OCP must be conducted to gain a clearer understanding.

### 5.3. Immunosuppression

It is well-known that immunosuppressed individuals are at an increased risk of developing many cancers, such as colorectal, liver, and stomach cancers [67]. These high-risk patients include but are not exclusive to those with immunodeficiency disorders and organ transplantation [68]. Patients with a cervix who are taking immunosuppressive medication post-organ transplantation are at an elevated risk of cervical cancer due to their vulnerability to HPV infections, driving the mechanism of oncogenesis [69]. A 2009 study from the Netherlands reported that renal transplant recipients were at a two-to-six-fold increase of CIN and a three-fold increased risk of cervical carcinoma compared to the immunocompetent population [70]. These findings were also supported by recent studies conducted in Brazil and China [71,72]. Furthermore, women living with human immunodeficiency virus (HIV) are also thought to be at risk due to the inability to clear the HPV infection effectively [73]. There is evidence that HIV-infected individuals are significantly more vulnerable to presenting with abnormal cervical cytology compared to non-HIV individuals [74]. Furthermore, an association between increased susceptibility to invasive cervical cancers with diminishing CD4 count has been observed [75]. Therefore, these high-risk patients should be warned of possible complications of immunosuppression and be provided vigilant screening with more regular follow-ups compared with the general population. This is supported by NICE guidelines, which offer screening within a year of transplantation, and for HIV-positive individuals, cervical screening, or colposcopy is offered at diagnosis and annually thereafter [76].

### 5.4. Sexual Behavioural Factors

Various aspects of sexual behaviours have been associated with increased cervical cancer risk, due to a higher chance of acquiring HPV infection [77]. Firstly, early age of sexual intercourse is a known risk factor, with one study reporting a 2.4-fold increase in developing invasive cervical cancer in girls aged 16 or younger [31]. Similarly, a large epidemiological study revealed that first sexual intercourse aged 14 or below compared to 25 and above reported a relative risk of 3.52 [77]. The current age of consent for sexual intercourse in the UK is 16 years old, therefore, offering the HPV vaccination before this age can provide adequate protection against the infection [78]. Furthermore, a meta-analysis investigated the significance of having multiple sexual partners in the development of cervical malignancies, which found that this was an independent risk factor, regardless of HPV status [32]. In contrast, a recent study did not find correlation between sexual behavior and HPV infection. As such, the interpretation of HPV-related head and neck cancer as a sexually transmitted disease should be treated with caution [79].

## 6. Current NHS Programmes—HPV Vaccination

The NHS vaccination programme currently offers the first HPV vaccine dose to children aged 12 to 13 years, with their second dose 6 to 24 months later [80]. These vaccinations are provided free of charge by the NHS for girls born after 1 September 1991 and boys born after 1 September 2006 [80]. Previously, there was a longstanding vaccination programme exclusively for girls until its extension to both sexes in September 2019 [80]. This was due to beliefs of herd immunity and indirect protection for boys against the HPV infection [81]. This does not equally safeguard men who have sex with men (MSM), and this population was, therefore, still unprotected from HPV-associated cancer risk [81]. Since early 2018, the NHS offers eligible MSM, who are 45 years old and under, free HPV vaccinations [80]. If an individual is eligible for the HPV vaccine, but does not opt for vaccination when offered, it remains available on the NHS at no cost to the individual until their 25th birthday [80]. It is important to note that three doses of the vaccine are recommended for immunocompromised persons and those who received their first dose at 15 years of age or older [82].

As shown in Figure 1, Cervavix, Gardasil, and Gardasil-9 are the three HPV vaccines approved by the Food and Drug Administration (FDA), designed with L1 VLPs to induce neutralising antibodies against the capsid protein [45,46,79,83,84]. The nine-valent HPV vaccine (Gardasil-9) is the current vaccine of choice and covers nine types of HPV: 6, 11, 16, 18, 31, 33, 45, 52, and 58 [85,86]. It will be implemented throughout 2022 as stock of the previous quadrivalent Gardasil vaccine becomes exhausted [87]. Gardasil-9 is a non-infectious recombinant vaccine that contains purified L1 proteins [88]. Studies report that vaccination protects against HPV infection for at least ten years, with some estimating longer lasting protection, although the long-term efficacy of Gardasil-9 is yet to be confirmed [89,90]. Research using simulation models found hypothetical evidence that without regular cervical screening, Gardasil-9 vaccination decreases the lifetime risk of cervical cancer and mortality by seven-fold [91]. It is important to note that despite being deemed to be effective in reducing cervical cancer by up to 88%, the vaccine does not protect against all HPV types culpable of cervical cancer development [92]. Consequently, it remains imperative that individuals who enrol in the immunisation programme, also receive regular cervical screening.

## 7. Current NHS Programmes—Cervical Cancer Screening

The NHS cervical screening (smear test) programme was first introduced in 1988 in the UK, which screened women aged 20 to 64 years of age every 3 to 5 years [93,94]. Between 1988 and 1992, the UK observed a striking reduction in mortality and an increased coverage [95]. However, due to evidence that screening women from age 20 had little or no effect on cancer risk, the NHS announced that women aged between 25 to 49 years were to be screened three-yearly, whilst women aged 50 to 64 years old would be invited to be screened every five years [15,96]. Northern Ireland has since adopted the same screening method, whilst Wales and Scotland have decreased the testing frequency to five-yearly for all eligible age groups [97,98]. The updated screening programme continues to be successful in decreasing incidence rates and mortality regarding cervical cancer [99].

Currently, the primary aim of the screening programme is to identify the HPV infection, thus, preventing cervical cancer [16]. Various randomised controlled trials revealed that a cervical screening test detecting hrHPV DNA had greater sensitivity in recognising CIN than a cytology-based screening method, thus, providing significantly more protection against cervical cancer [100,101,102]. Those who are informed of an abnormal result that are highly suggestive of CIN are referred to cytology to look for dyskaryosis. If present, those individuals will be invited for another screening in 12 months’ time [103]. However, if dyskaryosis is reported on cytology, women will be offered a colposcopy, to ensure a further detailed examination of the cervix [104]. Subsequently, women with severe dyskaryosis may be offered large loop excision of the cervical transformation zone (LLETZ) to remove abnormal cells, a procedure that has been hugely beneficial in reducing the overall mortality rate [105].

Data were collected from NHS Digital, where outcomes of the cervical screening programme are published annually. This includes data from the call and recall system that, according to the NHS, was minimally disrupted by the coronavirus pandemic, as it is shown in Figure 2 [106]. The date range for data collection for each year was from the 1st of April to the 31st of March of the following year [106]. ‘Screened from invitation’ measures screening uptake within a specific year and provides an indication of short-term screening uptake, whereas coverage data represent a summary of screening uptake over a longer period. Age-appropriate coverage was calculated through the number of women in the 25 to 49 and 50 to 64 years age group, who, within the last 3.5 and 5.5 years, respectively, have had an adequate screening test as a percentage of the eligible population.

In the pre-pandemic era, both the percentage of patients screened from invitation and the percentage of those with age-appropriate coverage trend towards a decrease over time. From April 2010 to March 2018, the number of patients screened from invitation decreased by 6% and age-appropriate coverage decreased by 4%. Age-appropriate coverage declined for four consecutive years during this pre-pandemic period. From April 2018 to March 2019, patients screened from invitation astronomically increased by 6% compared to the previous year. Despite how positive this may appear, it is still a 0.7% decrease from screening uptake in 2012/2013.

Due to COVID-19 measures, attendance for screening decreased in 2020. This does not entirely explain such an astronomical drop in screening from invitation for the period April 2019 to March 2020, as coronavirus measures only began to be implemented in March 2020, with the eleven previous months available for screening. The decline in screening uptake continued until March 2021, but an increase in screening uptake has since been observed. Age-appropriate coverage in the UK has trended towards an overall decline since April 2010.

## 8. The Effect of Coronavirus Pandemic on Cervical Cancer Screening Uptake

On the 16 March 2020, the UK government introduced its first coronavirus restrictions, where the Prime Minister announced for “everyone to stop non-essential contact and travel” [107]. General practice (GP) staff agreed that a transition towards remote consulting was essential to reduce the risk of COVID-19 infection [108]. The pandemic rapidly changed the landscape of healthcare, whereby 90% of GP consultations were conducted remotely as of April 2020, compared to 31% in the year prior [108]. Despite the positive aspects of a virtual triage policy during the pandemic, the unfortunate consequence was the risk of technological exclusion due to lack of technology and digital literacy [109]. This digital divide extends not only to access for knowledge, services, and goods but also to information and communication. In the UK, the number of internet non-users has declined, although in 2018, 5.3 million adults identified as such, and this must be improved in accordance with the United Nations sustainable development goals [110].

Furthermore, from April to July 2020, fewer consultations took place compared to the same period in 2019 [108]. The decrease in consultations is speculated to be due to staff sickness or shortages, reduction in treatment capacity due to hospital service limitations, reduced screening and investigation services, and screening or healthcare non-attendance due to concerns of contracting COVID-19 [111,112]. This resulted in GPs and nurses focusing on elderly patients, immunocompromised patients, and individuals with a mental health condition [108]. This burden is important to consider as screening programmes do not diagnose all cervical cancer cases, with some patients instead presenting to their GP with symptoms, to be given a 2 week wait (2 ww) referral to secondary care. The 2 ww referral pathway is thought to contribute to 22.1% of cervical cancer diagnoses [113]. Large backlogs of patients from national lockdown placed tremendous pressure on secondary care, resulting in 84% fewer 2 ww referrals being made [113]. A modelling study in the UK found that delays in presentation via the 2 ww pathway, with an average delay of two months per patient, would result in 22 additional lives and 559 life-years lost from cervical cancer [113].

## 9. Socioeconomic Factors on Cervical Cancer Screening Uptake

Concerningly, studies consistently find that screening coverage is worse among individuals from more deprived socioeconomic backgrounds [114]. HPV status is the primary screening method in the UK, although significant variations in response to testing HPV-positive have been studied across different socioeconomic backgrounds, which may affect future reattendance [115]. A large population-based survey investigated psychosocial responses, including shame, stigma, and anxiety among HPV-positive patients [115]. Primary education level, religion, age under thirty years old, and unemployment were all associated with adverse outcomes [115]. Educational level has been consistently associated with lower cervical screening coverage in several national surveys [116,117,118]. It is essential to minimise any possible adverse psychological effect of HPV screening, as this could negatively impact screening uptake. Potentially, the development of specific messages around HPV infection and screening could be tailored for particular subgroups of women to address the variety of concerns that may affect future non-attendance.

Incidence rates of cervical cancer in England were reported to be 65% higher in the most deprived areas compared with the least, with lifestyle factors such as cigarette smoking, increasing the likeliness of death by 21% compared to non-smoker patients [3,119]. Smoking rates are higher among deprived socioeconomic status individuals, and more specifically, characteristics such as income, housing, car availability, lone parenting, and neighbourhood deprivation significantly affect smoking prevalence [120]. Furthermore, lower socioeconomic status is associated with sexual behaviour, such as unprotected sexual intercourse with multiple partners, which increases HPV transmission risk [121]. However, it is important to note that sexual behaviour is an individual factor and develops under strong influence of cultural and other influences [121]. There also remains the financial burden for individuals who fall outside of the eligibility criteria for HPV vaccination on the NHS. In the private sector, as of January 2023, two doses would cost around £329 whilst three doses would cost £469 [122,123]. This is a tremendous cost to an individual, who in the context of a cost-of-living crisis in the UK, may not identify HPV vaccination as a priority and may experience future health complications due to financial limitations.

## 10. Future Direction

The NHS cervical screening and HPV vaccination programmes have shown to reduce incidence rates and provide adequate protection against cervical cancer [93,124]. The NHS healthcare system is free at the point of delivery, but health inequalities persist despite cost not being an obstruction to care [125]. Further work is needed in improving the uptake of these programmes in the UK, especially in individuals of lower socioeconomic status. Various studies from non-UK countries, including Denmark and the US, have reported remarkable findings on the impact of HPV home testing kits on screening participation, particularly among lower socioeconomic groups [126,127,128]. Furthermore, a randomised controlled trial in Italy observed a 40% increase in compliance with HPV self-sampling kits and in Sweden, 87% of women in one study preferred home testing compared to attending clinic-based appointments [129,130]. The WHO declared that self-sampling may help in reaching the global target of 70% screening coverage by the year 2030 [131]. The introduction of HPV home testing kits allows women who are non-attenders of clinic-based screening programmes to test for the HPV infection in the convenience of their own homes. This could be hugely successful in the UK, where inadequate cervical cancer screening uptake is an ongoing issue. Obstacles to engaging in the programme could be due to practicality (inconvenient clinic hours and transport issues), personal factors of feeling embarrassment or fear, and social-cultural determinants, such educational factors, and cultural beliefs [132]. Currently, there is no formal HPV home testing programme for under-screened women in the UK, but NHS England are trialling its first home testing kit initiative across 166 primary care centres in areas with particularly low attendances [133]. The official enrolment of home testing kits in the UK may significantly increase participation and uptake by overcoming these previously mentioned barriers. It should also be stressed to parents and girls that frequent cervical cancer screening should be completed despite the HPV vaccination, as the vaccine does not protect against all HPV types.

The uptake of the HPV vaccination programme has also been observed to be lower than expected across England, which may be due to several factors [134]. A large qualitative study across the UK, Germany, France, and Italy was conducted to determine the attitudes towards the HPV vaccination and the degree of acceptability among parents [135]. The study reported 75% of parents were in favour of the vaccine, regardless of gender, with the UK observing the highest compliance with the national immunisation programme compared with the other countries [135]. However, parents’ reasons to reject the vaccination programme included fear of unknown side-effects of vaccines in general and having inadequate knowledge of the HPV vaccine or HPV-related diseases [136]. Another qualitative survey in Sweden revealed that vaccine hesitancy, particularly among parents of boys, was due to low awareness of the benefits of the HPV vaccination of boys against a predominantly female disease [136]. These studies suggest that the lack of information or misinformation amongst the general population must be addressed. As most boys and girls receive their vaccination on school grounds in the UK, a possible method of achieving maximal uptake would be to focus on educating pupils alongside their parents or guardians, to inform them of the benefits and importance of the HPV vaccination and address any concerns they may have.

The issue of persistent low cervical cancer screening coverage in socially deprived areas must be tackled. Education in schools, primary care, and community centres is crucial in raising awareness of cervical cancer and HPV-related diseases, educating women on the symptoms of cervical cancer to improve early identification of the disease. Furthermore, raising awareness of strong risk factors associated with cervical cancer, including smoking and sexual behaviours, will be vital in the prevention of cervical cancer. In women who are frequent non-attenders to healthcare services, frequent reminders through text messages and phone calls from their registered GP may increase the compliance for screening and vaccination services. Transport, financial, and timing issues may contribute to non-attendance in lower socioeconomic areas, therefore, aiming to increase service accessibility by increasing more sexual health clinics and local screening service locations may be hugely beneficial in increasing uptake of services in these individuals. Lastly, the prevalence of cervical cancer may increase due to the delay of screening services during the COVID-19 pandemic [137]. A 2022 cohort study predicted an additional 919 cervical cancer cases due to the COVID-19 pandemic [138]. By increasing the capacity for healthcare appointments and surgeries post-pandemic, this could be vital in preparing for this predicted surge. By addressing potential reasons for reduced uptake of cervical cancer services and deliberating possible methods for achieving maximal compliance, the burden of cervical cancer may be significantly reduced in the UK.

## 11. Conclusions

Understanding of the HPV infection is crucial in the prevention, detection, and management of a significant majority of cervical cancers in the UK. Current programmes in the UK, which include the enrolment of the HPV vaccines, organised screening, and detailed colposcopy procedures, are highly beneficial in improving the prognosis and mortality of these patients. Prominent risk factors including cigarette smoking, immunosuppression, the OCP, and sexual behaviour have been identified as significant risk factors. Public health initiatives to educate the general population on these modifiable risk factors will help reduce the burden of disease. Furthermore, critiquing the possible reasons for reduced uptake of current NHS programmes and considering methods to ensure maximal compliance will be key in moving closer to eliminating cervical cancer in the UK.

## Figures and Tables

**Figure 1 pathogens-12-00298-f001:**
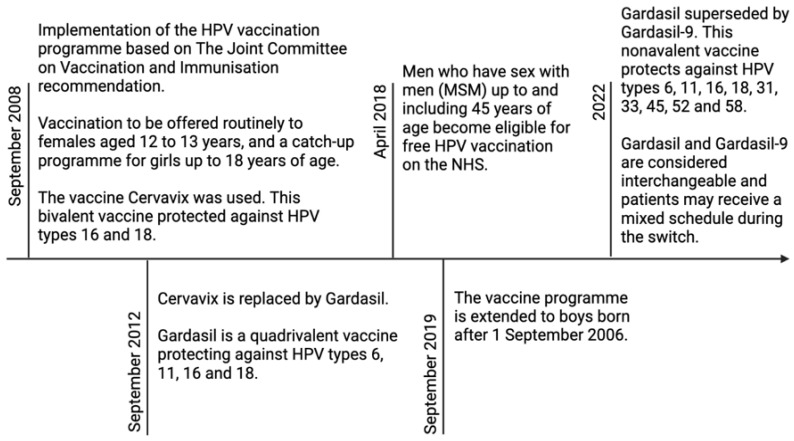
Timeline of HPV vaccination programme implementation and adjustments.

**Figure 2 pathogens-12-00298-f002:**
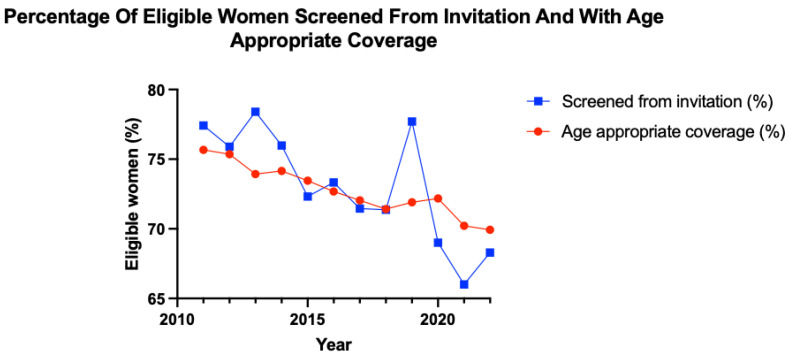
Graph demonstrating the changes in the percentage of women screened from invitation and percentage of women with age-appropriate coverage from April 2010 to March 2022.

## Data Availability

Not applicable.

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
