# Peer review of "HPV and Cervical Cancer: A Review of Epidemiology and Screening Uptake in the UK"

_pathogens, 2023, doi:10.3390/pathogens12020298_

Round 1

Reviewer 1 Report

In the manuscript entitled “HPV and Cervical Cancer: A Review of Epidemiology and Screening Uptake in the UK” the authors present significant information concerning the epidemiology of cervical cancer and current prevention programmes within the NHS. The manuscript is very informative and well-written. However, some points need to be addressed.

1. Line 140. The authors mention that “After infectious internalisation, seven early HPV proteins, E1 to E7, are released from the capsid and transported into the nucleus”. This is not correct as the entire viral DNA in transported into the nucleus and it remains as episome or it is integrated into the host chromosome. The entire sentence must be changed. J. Virol. 2013;87:6062–6072. doi: 10.1128/JVI.00330-13, Front. Oncol. 2019;9:355. doi: 10.3389/fonc.2019.00355, Virol. J. 2010;7:11. doi: 10.1186/1743-422X-7-11, Expert Rev Mol Med. 2021;23:e19. doi:10.1017/erm.2021.18

2. In section “HPV subtypes” the description of viral DNA structure would help the readers to better understand the biology of HPV. Moreover, it would be beneficial to mention more information concerning the phylogenetic classification of HPV genome as well as to describe in detail the members of HR and LR HPV genotypes, including the criteria that were used to classify the HPV genotypes in these two groups.

3. The authors report the nine-valent HPV vaccine (Gardasil-9) as the current vaccine of choice. A brief description of all the available HPV vaccines that FDA has approved are required to be mentioned. Moreover, details considering the design of the available vaccines need to be incorporated in the manuscript. Hum. Vaccines Immunother. 2019;15:146–155. doi: 10.1080/21645515.2018.1512453, Viruses. 2023, 15:141. doi:10.3390/v15010141, Viruses. 2022 Apr 5;14(4):757. doi: 10.3390/v14040757. PMID: 35458487

4. Finally, a brief description of previous data concerning the prevalence of different HPV genotypes in the UK is missing

Author Response

Dear Editor and Referee,

We are pleased to resubmit for publication the revised version of Pathogens-2204109 manuscript, entitled, ‘HPV and Cervical Cancer: A Review of Epidemiology and Screening Uptake in the UK’.

The reviewer kindly provided us with a great deal of guidance, regarding how to further improve the article. We hope that you agree with this revised manuscript. All the comments have been addressed, as shown in the revised version of the manuscript, along with point-by-point response to the reviewer's comments shown below.

All corresponding changes are in blue in the manuscript.

Referee 1:

General Comments:

In the manuscript entitled “HPV and Cervical Cancer: A Review of Epidemiology and Screening Uptake in the UK” the authors present significant information concerning the epidemiology of cervical cancer and current prevention programmes within the NHS. The manuscript is very informative and well-written. However, some points need to be addressed.”

Response:

We appreciate you taking the time to offer us your comments and insights related to the paper. Thank you for your positive encouragement and constructive feedback. We did our best to respond to your concerns and revised the manuscript as such.

Specific Comments:

  1. Line 140. The authors mention that “After infectious internalisation, seven early HPV proteins, E1 to E7, are released from the capsid and transported into the nucleus”. This is not correct as the entire viral DNA in transported into the nucleus and it remains as episome or it is integrated into the host chromosome. The entire sentence must be changed.”

Response:

Thank you for your comment. The sentence has now been removed and replaced by the correct information regarding HPV viral DNA (Lines 152-154). References amended accordingly (38 and 39).

  1. In section “HPV subtypes” the description of viral DNA structure would help the readers to better understand the biology of HPV. Moreover, it would be beneficial to mention more information concerning the phylogenetic classification of HPV genome as well as to describe in detail the members of HR and LR HPV genotypes, including the criteria that were used to classify the HPV genotypes in these two groups.”

Response:

Thank you for your comment. We have updated that section and include a description of the viral DNA structure in HPV, in order to understand how HPV can infect the host cell. The phylogenetic classification of the HPV genome has also been added Furthermore, both high risk and low risk HPV genotypes have been listed in full, explaining how they were classified. References amended accordingly.

  1. The authors report the nine-valent HPV vaccine (Gardasil-9) as the current vaccine of choice. A brief description of all the available HPV vaccines that FDA has approved are required to be mentioned. Moreover, details considering the design of the available vaccines need to be incorporated in the manuscript.”

Response:

Thank you for your comment. We agree that all available HPV vaccines that are FDA-approved should be mentioned. The three HPV vaccines approved by the FDA are Cervavix, Gardasil and Gardasil-9. In, Figure 1, we address all three vaccines with a brief description of their introduction into the UK vaccination programme and HPV subtypes targeted. Emphasis of their FDA approval and design has been added (Lines 269-271).

  1. Finally, a brief description of previous data concerning the prevalence of different HPV genotypes in the UK is missing.”

Response:

Thank you for this comment. We have now included significant data that found a high prevalence of high-risk HPV types 16 and 18 in young women in the UK, as well as incidence of other high-risk HPV types, including 31, 33, 45, 52 and 58 (Lines 129-132).

Reviewer 2 Report

This review basically comprehensively looks at the topic of "HPV and cancer of the cervix" in the context of incidence, screening and vaccination programs in addition considering the Covid pandemic. basically, I can recommend the publication of the paper as it is. However, if necessary, the following points could be considered: 1. In section 4, the topic of latent infection in the anogenital tract, which does not exist in the oropharynx, for example, could be better described. It is known that promiscuous MSM have increased HPV in the oral cavity, but also lose it more quickly. The reasons for the latter and latency are unclear. 2. also in section 4, it could be mentioned more strongly that HPV rely on the replication system of the human cell, as they do not have their own: accordingly, the intrinsically differentiated dormant cells must be induced to replicate again, which the viruses achieve with E6 and E7 activity. 3. Chapter 5.4 deals with sexual behavior. I would like to draw your attention to a separate paper in which we critically examine these aspects in great detail from the point of view of head and neck virology. The authors may want to consider whether they would like to include the criticism of overly simple correlations in their paper (  DOI: 10.1016/j.pvr.2020.100207)

The list of authors is ending with "and....6" instead of  and 6

Author Response

Dear Editor and Referee,

We are pleased to resubmit for publication the revised version of Pathogens-2204109 manuscript, entitled, ‘HPV and Cervical Cancer: A Review of Epidemiology and Screening Uptake in the UK’.

The reviewer kindly provided us with a great deal of guidance, regarding how to further improve the article. We hope that you agree with this revised manuscript. All the comments have been addressed, as shown in the revised version of the manuscript, along with point-by-point response to the reviewer's comments shown below.

All corresponding changes are in blue in the manuscript.

Referee 2:

General Comments:

This review basically comprehensively looks at the topic of "HPV and cancer of the cervix" in the context of incidence, screening and vaccination programs in addition considering the Covid pandemic. basically, I can recommend the publication of the paper as it is. However, if necessary, the following points could be considered:

Response:

We are grateful for your kind words about our paper and appreciate the opportunity to revise our work, given your expertise.

Specific Comment:

  1. 1. In section 4, the topic of latent infection in the anogenital tract, which does not exist in the oropharynx, for example, could be better described. It is known that promiscuous MSM have increased HPV in the oral cavity, but also lose it more quickly. The reasons for the latter and latency are unclear.”

Response:

Thank you for your comment, and we fully agree. However, we did not feel comfortable to incorporate in that section that promiscuous MSM have increased HPV in the oral cavity.

  1. 2. also in section 4, it could be mentioned more strongly that HPV rely on the replication system of the human cell, as they do not have their own: accordingly, the intrinsically differentiated dormant cells must be induced to replicate again, which the viruses achieve with E6 and E7 activity.”

Response:

Thank you for your comment, we acknowledge this request and added lines 159-162.

  1. 3. Chapter 5.4 deals with sexual behavior. I would like to draw your attention to a separate paper in which we critically examine these aspects in great detail from the point of view of head and neck virology. The authors may want to consider whether they would like to include the criticism of overly simple correlations in their paper ( DOI: 10.1016/j.pvr.2020.100207)”

Response:

Thank you for your comment. We had the opportunity to read with great interest of your article. We have added a comment along with the reference (lines 247-250, ref. 80)

The list of authors is ending with "and....6" instead of and 6

Response:

Revised accordingly; thank you.

Reviewer 3 Report

The review is well described about the current situation of the epidemiology of cervical cancer, HPV pathogenesis and transmission, and current programmes within the NHS for prevention and early identification of this disease, and concluded with detailed analysis into screening uptake, risk factors and future developments, and the importance in reducing the burden of cervical cancer in the UK.

The article is informative, educational, almost completed. The reviewer found it worth being published.

One request from the reviewer to the authors is to add a small comment about lymphedema after cervical cancer survivorship.

If possible, please add the comment after line 49-51 “It is the fourth most common malignancy~United Kingdom (UK).”, with referring the following articles, for example, “In addition, it is known that lower-limb lymphedema occurs at a constant rate after cervical cancer survivorship(Risk factors for lower-limb lymphedema after surgery for cervical cancer. Int J Clin Oncol. 2011 Jun;16(3):238-43.), and the lymphedema patients suffer from repeating cellulitis, delayed wound healing, as well as increasing volume of lower limb (Lymphovenous Anastomosis Aids Wound Healing in Lymphedema: Relationship Between Lymphedema and Delayed Wound Healing from a View of Immune Mechanisms. Adv Wound Care (New Rochelle). 2019 Jun 1;8(6):263-269.).”.

Author Response

Dear Editor and Referees,

We are pleased to resubmit for publication the revised version of Pathogens-2204109 manuscript, entitled, ‘HPV and Cervical Cancer: A Review of Epidemiology and Screening Uptake in the UK’.

The reviewers kindly provided us with a great deal of guidance, regarding how to further improve the article. We hope that you agree with this revised manuscript. All the comments have been addressed, as shown in the revised version of the manuscript, along with point-by-point response to the reviewers’ comments shown below.

All corresponding changes are in blue in the manuscript.

Referee 3:

General Comments:

“The review is well described about the current situation of the epidemiology of cervical cancer, HPV pathogenesis and transmission, and current programmes within the NHS for prevention and early identification of this disease, and concluded with detailed analysis into screening uptake, risk factors and future developments, and the importance in reducing the burden of cervical cancer in the UK.

The article is informative, educational, almost completed. The reviewer found it worth being published.”

Response:

Thank you for taking the time to review our manuscript. We are grateful for your kind words about our paper and appreciate the opportunity to revise our work for consideration for publication.

Specific Comment:

  1. “One request from the reviewer to the authors is to add a small comment about lymphedema after cervical cancer survivorship.

If possible, please add the comment after line 49-51 “It is the fourth most common malignancy~United Kingdom (UK).”, with referring the following articles, for example, “In addition, it is known that lower-limb lymphedema occurs at a constant rate after cervical cancer survivorship(Risk factors for lower-limb lymphedema after surgery for cervical cancer. Int J Clin Oncol. 2011 Jun;16(3):238-43.), and the lymphedema patients suffer from repeating cellulitis, delayed wound healing, as well as increasing volume of lower limb.”

Response:

Thank you for your comment, we acknowledge this request. Lymphoedema is an important and prevalent complication of cervical cancer treatment. The focus of our manuscript is on epidemiology and screening uptake and the discussion of cervical cancer treatment complications and survivorship is outside the scope of our study. We acknowledge the importance of such complications, although other complications such as bladder, bowel and sexual dysfunction and psychological effect must also, therefore, be discussed in accordance with your comment, all of which fall outside the remit of our manuscript.

Round 2

Reviewer 1 Report

The authors have improved the manuscript and addressed my comments. It is suitable for publication